# Informer-WGAN: High Missing Rate Time Series Imputation Based on Adversarial Training and a Self-Attention Mechanism

**Yufan Qian [1], Limei Tian [2], Baichen Zhai [2], Shufan Zhang [1] and Rui Wu [1,*]**

[1] Faculty of Computing, Harbin Institute of Technology, Harbin 150000, China; 20s103205@stu.hit.edu.cn (Y.Q.); 21s103149@stu.hit.edu.cn (S.Z.)
[2] Beijing Institute of Control Engineering, Beijing 100190, China; tian-maggie@bidri.com.cn (L.T.); baichen-zhai@bidri.com.cn (B.Z.)
[*] Correspondence: simple@hit.edu.cn; Tel.: +86-15846592891

**Abstract:** Missing observations in time series will distort the data characteristics, change the dataset expectations, high-order distances, and other statistics, and increase the difficulty of data analysis. Therefore, data imputation needs to be performed first. Generally, data imputation methods include statistical imputation, regression imputation, multiple imputation, and imputation based on machine learning methods. However, these methods currently have problems such as insufficient utilization of time characteristics, low imputation efficiency, and poor performance under high missing rates. In response to these problems, we propose the informer-WGAN, a network model based on adversarial training and a self-attention mechanism. With the help of the discriminator network and the random missing rate training method, the informer-WGAN can efficiently solve the problem of multidimensional time series imputation. According to the experimental results under different missing rates, the informer-WGAN model achieves better imputation results than the original informer on two datasets. Our model also shows excellent performance on time series imputation of the key parameters of a spacecraft control moment gyroscope (CMG).

**Keywords:** informer; GAN; high missing rate; time series imputation

## 1. Introduction

Recently, with the continuous acceleration of industrialization, industrial sensors are used in all walks of life to obtain various data and are widely used in many fields such as weather forecasting, medical treatment, power statistics, and aerospace telemetry [1]. A time series is the data collected at a certain frequency of industrial physical quantities monitored by sensors, since multiple sensors in an industrial scene are often correlated and sampled from multiple industrial sensors to obtain a multidimensional time series. However, due to the problem of the hardware device, the multidimensional time series in the actual scene may have the problem of missing values. Not only that, but for some fields, such as weather forecasting and stock forecasting, multidimensional time series forecasting is also a very valuable research problem [2], both of which can be defined as imputation problems for missing data. For imputation of multidimensional time series, sufficient analysis of the time series is required.

In order to deal with the imputation problem of multidimensional time series, researchers have provided many traditional algorithms, forming the initial solutions. For example, the autoregressive integrated moving average (ARIMA) treats the imputation of time series as the process of calculating the unknown dependent variable from the known independent variables [3]. Schafter applied the Markov chain Monte Carlo method (MCMC method) to data imputation for processing data with multivariate arbitrary missing patterns [4]. However, these methods often require some prior knowledge, such as the difference order and autocorrelation lag order of the ARIMA [5]. At the same time, for complex multidimensional time series with high missing rates, due to the incomplete

utilization of time series characteristics, the performance of these methods is generally poor [6].

In recent years, many machine learning methods have been used for multidimensional time series imputation problems. Rhaman et al. used the EM algorithm to combine the decision tree and build a decision forest. After classifying and merging the data, the EM algorithm was used to implement data imputation. Through experiments on six public datasets with missing values and three artificially made missing value datasets, as well as a comparison with the DMI and SiMI algorithms, the effectiveness of this method for the imputation of missing data in attribute classes was verified [7]. In 2015, Folguera et al. proposed applying a self-organizing map neural network (SOM) to the data imputation task to predict the physical and chemical parameters of water samples. An SOM uses a nonlinear method to automatically fill in missing values in the data matrix. Folguera verified the SOM method in data imputation through experiments on surface water data (17–39% missing rate) sampled from the Reconquista River in Argentina. Regarding task effectiveness [8], Langkvist provided a review of the methods related to building prediction models for time series through deep learning. The two models that perform well in sequence modeling problems are based on self-attention mechanism models and recurrent neural network-based models [9]. DeepAR, proposed by Salinas et al., utilizes an encoder-decoder structure combined with an autoregressive RNN model for time series prediction [10]. These machine learning methods have all conducted valuable explorations for the multidimensional time series imputation problem. Inspired by Langkvist's review, we adopted a network model based on the self-attention mechanism to impute time series.

The main contributions of this paper include the following two aspects:

- We propose a model for multidimensional time series imputation, which is based on the WGAN-GP framework and uses the informer part of the network structure to form the generator and the discriminator. The networks proposed outperform the original informer model and the GAN-based AST model in both real-world datasets.
- We propose a random missing rate training method for the time series imputation problem which improves the accuracy of the data imputation model for time series imputation with different missing rates.

## 2. Related Work

### 2.1. Time Series Imputation

The data imputation method is to solve the problem of missing data, which refers to the phenomenon where the dataset is incomplete due to a missing value for one or more attributes. Missing data will not only distort the data characteristics and increase the difficulty of analysis but also change the statistics such as expectations and high-order distances, causing serious deviations in the analysis methods based on statistical models. For the imputation of missing data in time series, many researchers have proposed solutions before. The ARIMA model proposed by Box-Jenkins uses the autocorrelation of the time series lag k order to construct an autoregressive model [3] for time series forecasting. To predict the time series which exhibit long-range dependence or persistence in their observations, Javier E. Contreras-Reyes compared several autoregressive fractionally integrated moving average (ARFIMA) models [5]. Augustine Pwasong and Saratha Sathasivam [11] combined the ARFIMA model and a layer recurrent neural network (LRNN) to form a hybrid forecasting model. The ARIMA-LRNN model achieved a very robust and dependable result. Liu et al. combined the online learning algorithm with the ARIMA algorithm and proposed the online ARIMA algorithm. The online ARIMA algorithm is mainly used to solve the difficult problems of parameter estimation in the traditional ARIMA algorithm, such as lag order and interference terms. Liu reduced the ARIMA model to an online parameter optimization problem and verified through experiments that the online ARIMA algorithm performed better than the traditional ARIMA algorithm [12]. The network model of time series imputation based on an RNN is also a major research direction [13]. DeepAR was proposed by Salinas et al. by using the encoder-decoder structure combined with

the autoregressive RNN model for time series prediction [10]. LSTNet is another RNN-based autoregressive model that utilizes long short-term memory (LSTM) for time series prediction [14]. However, due to the natural disadvantages of the RNN network model, parallel operations cannot be used to improve the imputation efficiency, especially when the missing rate is high, at which point it shows poor performance.

## 2.2. Generative Adversarial Networks

A generative adversarial neural network is a kind of generative deep neural network proposed by Ian et al. in 2014 [15] through the adversarial training of a generator and discriminator to learn the distribution characteristics of the original dataset. Generative adversarial networks have achieved great success in computer vision and natural language processing in recent years, and at the same time, several studies have also been applied to generate sequence data [16]. After using the method of machine learning to complete the data imputation, we need to train the model by minimizing the likelihood function, but the likelihood function is limited to the description of the sequence characteristics, meaning that the discriminator is essentially fitting a better likelihood function to identify the complete sequence after imputation.

In 2019, Yoon et al. used a GAN and unsupervised learning to generate time series data and proposed TimeGAN, which achieved the best results for time series generation tasks. This model is not directly applicable to data imputation, as it was only used to model historical time series data [17]. Luo et al. proposed a method using E2EGAN for multidimensional time series imputation using a gate recurrent unit (GRU) as a generator and a discriminator network and achieved the best results on multiple real-world datasets. At the same time, the E2EGAN model is superior to the two-stage method in terms of time consumption. Combining the idea of a GAN and the self-attention mechanism, Wu et al. proposed an adversarial sparse transformer (AST) model, which achieved optimal results on multiple real-world datasets. The AST model uses a sparse attention encoder and decoder as the generator network and uses a linear fully connected layer as the discriminator network, which improves the imputation performance of the model compared with the RNN-based algorithm. At the same time, Wu also proved through ablation experiments that adversarial training helps to obtain better data imputation models. However, the AST model adopts the step-by-step generation method, which has low imputation efficiency and has the problem of error accumulation [6].

## 2.3. Attention Mechanism

The attention mechanism has been shining in the field of time series processing and prediction since it was proposed in 2017 [18], as it solves the problem that the RNN model can by not only predicting in a single step in time series prediction but can also improve the efficiency of the time series processing. Especially in long sequence prediction, the attention mechanism has a tendency to replace RNN methods such as LSTM. Some recent works also used sparse attention [19,20] based on the attention mechanism. Due to the softmax mechanism, the traditional attention mechanism gives an attention score to each unit in the embedding even if the correlation is extremely low, resulting in distraction and reduced computing efficiency. The sparse attention mechanism sets the irrelevant units to zero, which will make the model more focused. Experiments have shown that the sparse attention-based model is effective for time series imputation tasks on some datasets. In 2021, the informer proposed by Zhou et al. used convolution and pooling operations to perform knowledge distillation on the embedded data, and they were combined with the prob attention (i.e., sparse attention) to improve the performance of the model [21].

## 3. Materials and Methods

### 3.1. Materials

Dataset

We used three public real-world datasets (electricity, ETTh1, and CMG) for our evaluation. The electricity dataset (Table 1), which consists of hourly electricity consumption data collected from 370 customers, was used to evaluate the AST model. The ETTh1 dataset is hourly power load index data collected from several counties in China, and it was used for the informer model. The cmg dataset is the real telemetry time series data of the key parameters of a spacecraft's control moment gyro (CMG).

**Table 1.** Dataset data, where F represents the sampling frequency, L represents the number of samples, and D represents the data dimension.

| Dataset | F | L | D |
|---|---|---|---|
| electricity | hourly | 32,304 | 370 |
| ETTh1 | hourly | 17,420 | 6 |

### 3.2. Informer-WGAN Model

In this section, we will present the details of the proposed time series imputation method. The generative adversarial network (GAN) consists of a generator network and a discriminator network. The generator network takes low-dimensional random noise as its input and outputs high-dimensional fake data according to different tasks. The discriminator network takes fake data and real data as the input and outputs a number in the range of (0, 1) as the evaluation of the input data. The discriminator network will try to score real data close to 1 and fake data close to 0, while the generator network will try to generate fake data that makes the discriminator network give a high score. We would obtain an excellent generator that generates realistic data taking advantage of adversarial training of the generator and discriminator. In order to avoid the mode collapse problem in GAN training, we adopted the WGAN-GP model (please see Figure 1) as the neural network framework while making some modifications to the original loss function.

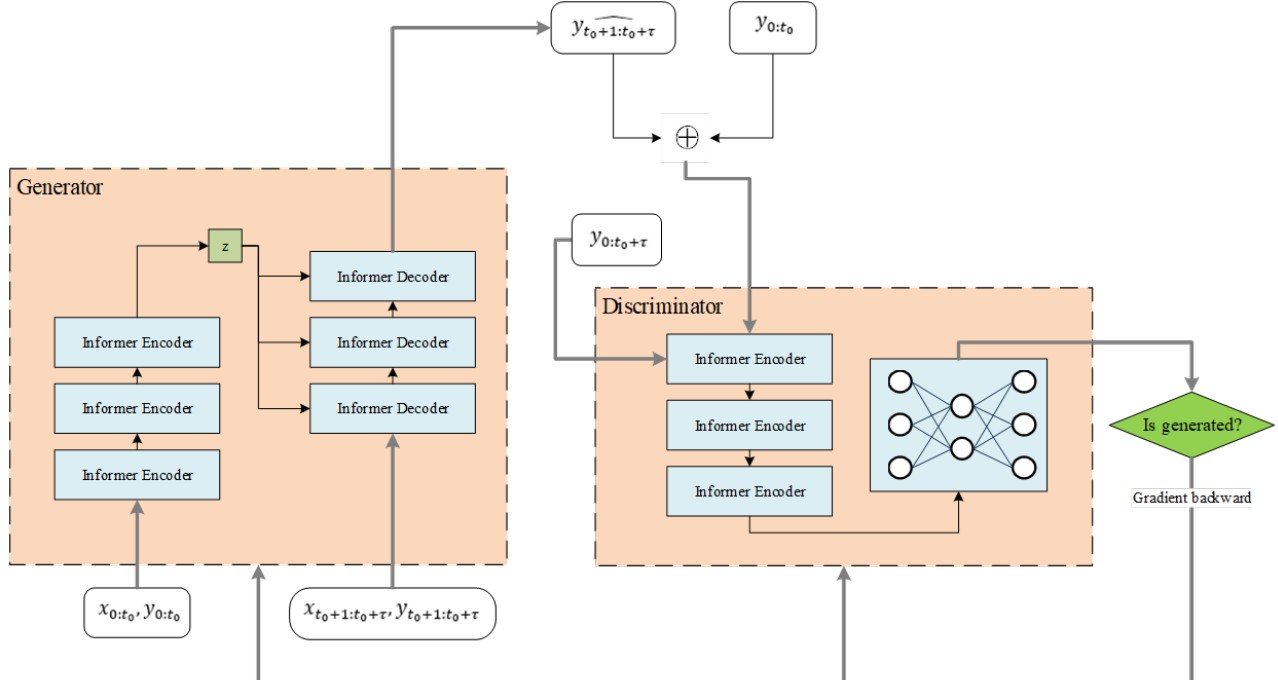

**Figure 1.** Informer-WGAN structure.

Our network structure refers to the adversarial sparse transformer (AST) model, and we redesigned and implemented the generator network and the discriminator network based on it.

### 3.2.1. Problem Definition

The problem is multidimensional time series imputation, where $d$ represents the dimension of the time series, $t$ represents the length of the time series, and thus the time series is defined as $x \in \mathbf{R}^{d \times n}$. According to the time feature of the sequence, we generated a covariate matrix through the sequence timestamp information $cov, cov \in \mathbf{R}^{d_{cov} \times n}$, where $d_{cov}$ describes the information of dimensions such as *month*, *day*, *hour*, and *weekday* in the sequence timestamp. We decided the specific $d_{cov}$ according to the feature of the actual time series, such as the period and trend.

### 3.2.2. Generator

In our proposed generator network Figure 2, the informer is mainly used as the generator network. Different from the specified missing length and token length of the original informer, the random missing rate and data window are used to calculate the missing length, which increases the robustness of the network for imputation under different missing rates of time series.

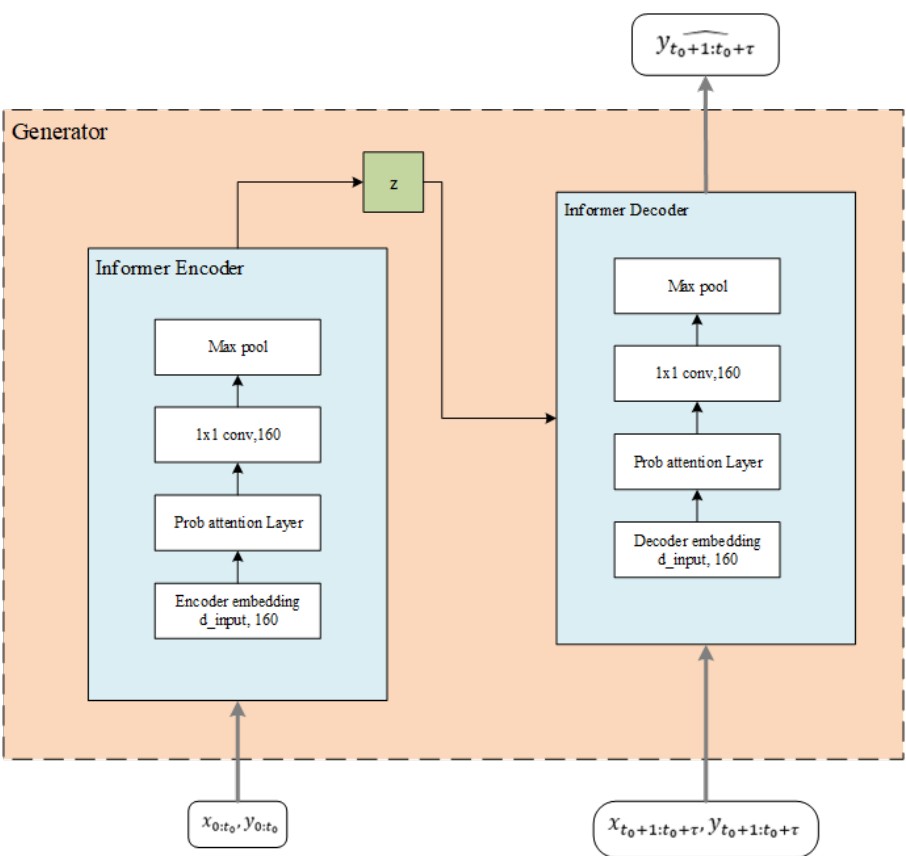

**Figure 2.** Generator structure.

First, according to the missing rate $r$, the data window size $L$, and the formulas $t_0 = L * (1 - r)/100$ and $\tau = L * r/100$, we calculated the known time series and the unknown part. The encoder part receives $x_{0:t_0}$ and $cov_{0:t_0}$ as inputs, where $x_{0:t_0}$ is the real value of the time series from 0 to $t_0$ and $cov_{0:t_0}$ is the part of the covariate matrix from 0 to $t_0$, calculated according to the time characteristics of the time series mentioned above. After the operation of n encoder parts, the network outputs the memory module $z$. The input of the decoder includes the output of the encoder $z$, *dec_input*, and

$cov_{t_0+1-token\_len:t_0+\tau}$ to participate in the self-attention operation of the decoder part, where $dec\_input = concat(x_{t_0+1-token\_len:x_{t_0}}, 0_{t_0+1:t_0+\tau})$. After n decoders, the generator outputs a fake sequence $\hat{x}_{t_0+1-token\_len:t_0+\tau}$, which is spliced with the real value of the time series from 0 to $t_0 + \tau$, to obtain the full time series forecast from 0 to $t_0 + \tau$ $\hat{x}$.

The loss function of the generator is a bit different from that of the original WGAN-GP:

$$L_G = \alpha * D(\hat{x}) + Q_{50}(G(z), x_{t_0+1:t_0+\tau}) \tag{1}$$

Since the discriminator network evaluates the entire complete sequence, we paid more attention to the sequence part of the actual imputation, which was achieved by the *quantile*-50 loss.

### 3.2.3. Discriminator

Different from the AST model, which uses a multi-layer fully connected layer as the discriminator, the discriminator network Figure 3 we propose uses the encoder part of the informer model to extract the features of the time series. The time series features are extracted through a multi-layer encoder, and the discrimination results are output through a multi-layer fully connected layer.

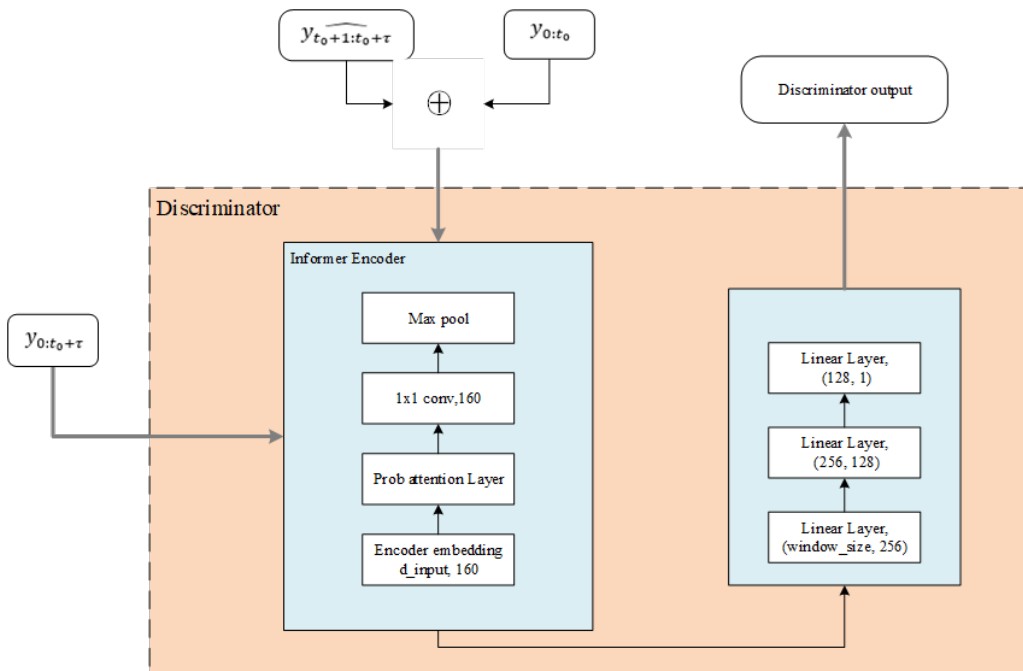

**Figure 3.** Discriminator structure.

The discriminator is used to distinguish the real value $x$ of the time series from the fake sequence $\hat{x}$ generated by the generator. The goal is to give a high score to the real value of the sequence and a low score to the fake sequence, so the loss function is as follows:

$$L_D = D(\hat{x}) - D(x) + gradient\_penalty \tag{2}$$

### 3.2.4. Random Training

According to the problem definition, in order to enhance the robustness of the model and enable the model to generate high-quality imputation data at various missing rates, especially at high missing rates, we propose a network training process based on random missing rates. The process is as follows Algorithms 1:

---

**Algorithm 1** GAN training process based on random missing rate.

---

1: **for** each training iteration **do**
2:     randomly sample $[X,Y]$ from dataset by $window\_size$
3:     set $missing\_rate = random(10, 80)$
4:     set $missing\_len = window\_size * missing\_rate // 100$
5:     compute $decoder\_input$ by $missing\_rate$ and $token\_len$
6:     compute     $Y_{fake}$     by     $Generator$     $G$     with     $Y_{fake}$     $=$
    $G(y_{known}, cov_{known}, decoder\_input, cov_{missing})$
7:     update $Generator$ and $Discriminator$
8: **end for**

---

## 4. Results

### 4.1. Evaluation Indicators

There are generally two types of evaluation indicators for time series imputation results. The first is the evaluation indicators using regression models, such as MAE and MSE. The other is to use correlation coefficients, such as cosine similarity, to calculate time series as vectors. We evaluated the time series imputation results with reference to the root mean square error (RMSE) and Pearson correlation coefficient (PCC). The RMSE and Pearson correlation coefficient formulas are as follows:

$$RMSE(\hat{x}, x) = \sqrt{\frac{1}{m} \sum_{i=1}^{m} (\hat{x}_i - x_i)} \tag{3}$$

$$PCCs(\hat{x}, x) = \frac{N \sum \hat{x}_i x_i - \sum \hat{x}_i \sum x_i}{\sqrt{N \sum \hat{x}_i^2 - (\sum \hat{x}_i)^2} \sqrt{N \sum x_i^2 - (\sum x_i)^2}} \tag{4}$$

The smaller the root mean square error, the better, and the larger the Pearson correlation coefficient, the better. Therefore, we present the comprehensive evaluation indicators as follows:

$$p\_RMSE = \frac{RMSE(\hat{x}, x)}{PCCs} \tag{5}$$

### 4.2. Implementation Details

We used an Nvidia RTX3080 to train the model for 100 epochs on the dataset for experimentation and result comparison. The discriminator network was trained using the RMSProp algorithm, and the generator network was trained using the Adam algorithm. For both datasets, 80% was used as the training set, 10% was used as the validation set, and 10% was used as the test set.

### 4.3. Performance Comparison

Figure 4 shows the actual imputation effect of our proposed model on the electricity dataset. The upper half is the time series imputation value, the lower half is the time series real value, and the red dots describe the missing data. Under the high missing rate, our model can still complete high-quality time series imputation, which fully demonstrates the performance of the informer-WGAN model. We also applied the informer-WGAN model on the CMG dataset, and Figure 5 shows the imputation results on the CMG dataset at the missing rates of 20% and 50%. Our proposed method revealed trends in the CMG time series. The pRMSE was 0.583 at the 20% missing rate and still reached 0.729 at the 80% missing rate. This shows that informer-WGAN is suitable for CMG time series imputation.

Figures 6 and 7 show the frequency hist of the imputation results and real-time series on the electricity dataset and ETTh1 dataset. The imputation series demonstrates obvious similarity with the real-time series. The frequency hist shows the effect of our imputation model.

Figure 8 shows the actual results of different imputation methods on the ETTh1 dataset at missing rates of 50% and 80%. Obviously, our informer-WGAN performed better than the other imputation methods such as ARIMA and informer, especially when the missing rate was high.

In order to show the actual effect of the model proposed, we chose and compared the traditional data imputation methods, including the ARIMA, the AST model, and the original informer, to compare the data imputation of each model under different missing rates. We also modified the original AST model and compared its effect, and we named it as ST-WGAN. The supplementary results were quantitatively evaluated by the p_RMSE value proposed in the previous section.

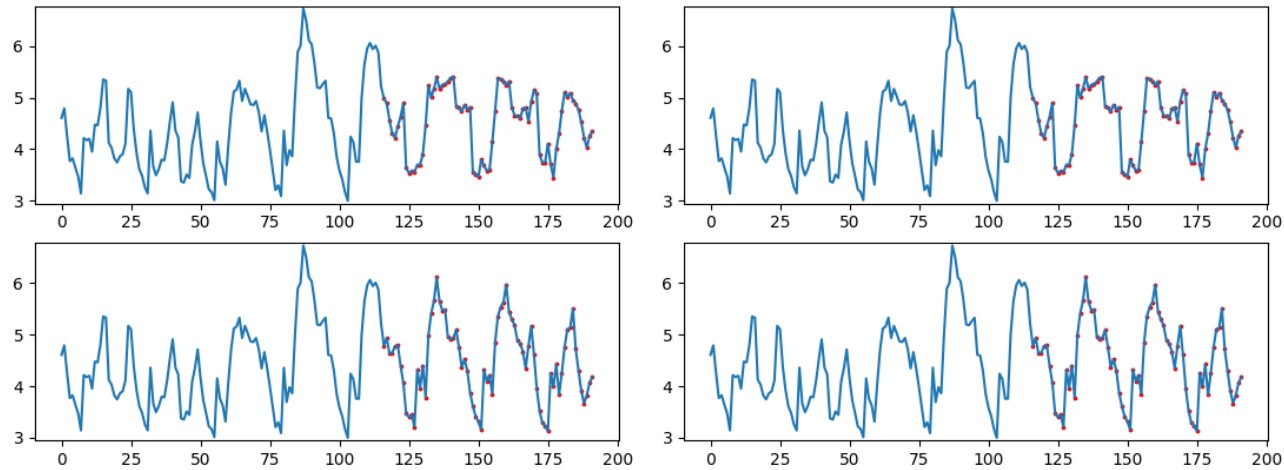

**Figure 4.** Imputation results of informer-WGAN method on electricity dataset at missing rates of 20% and 50%.

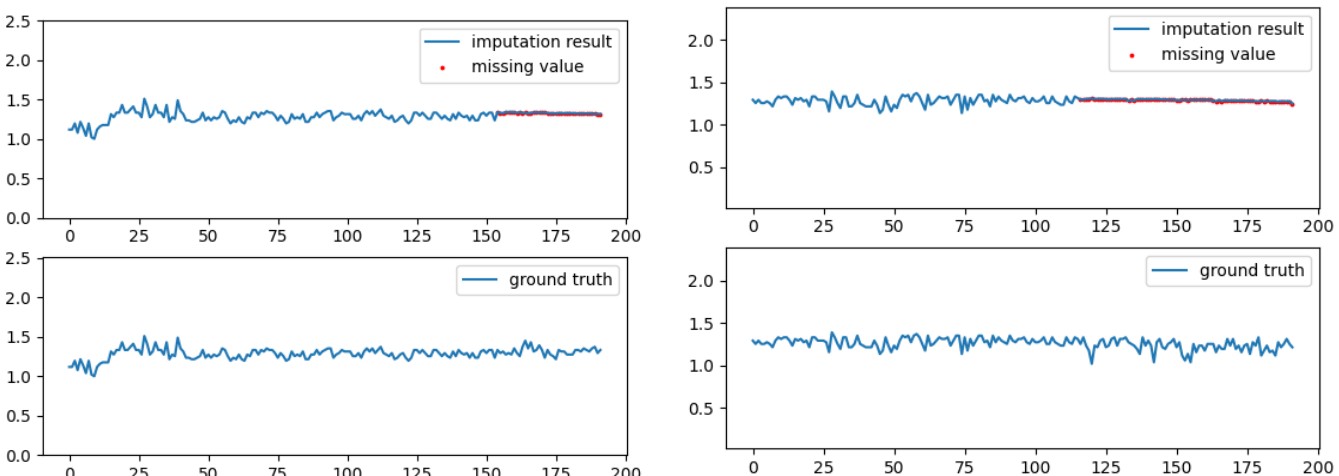

**Figure 5.** Imputation results of informer-WGAN method on CMG dataset at missing rates of 20% and 50%.

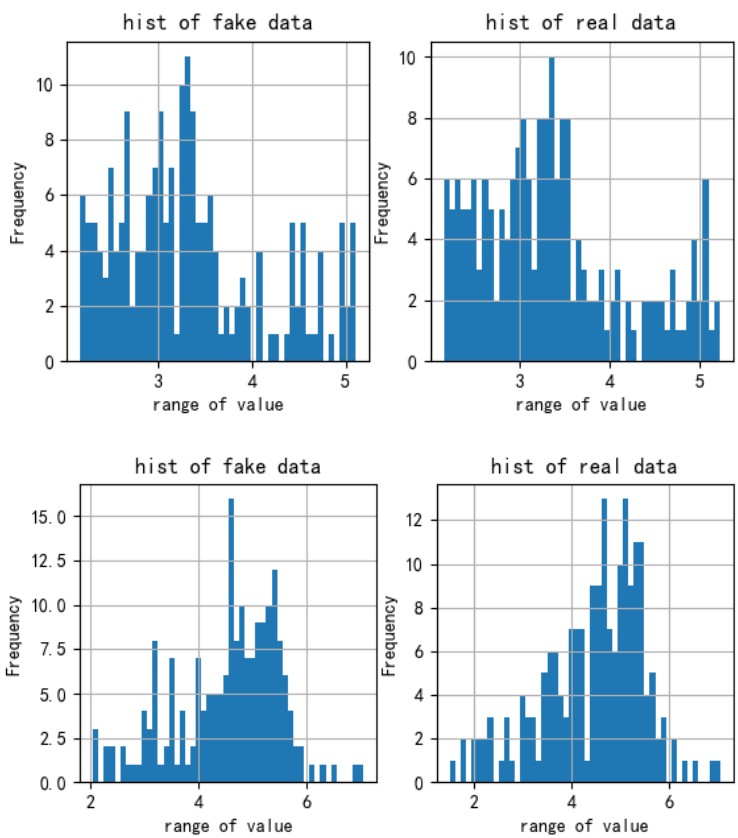

**Figure 6.** Frequency hist of imputation results and real-time series on electrity dataset.

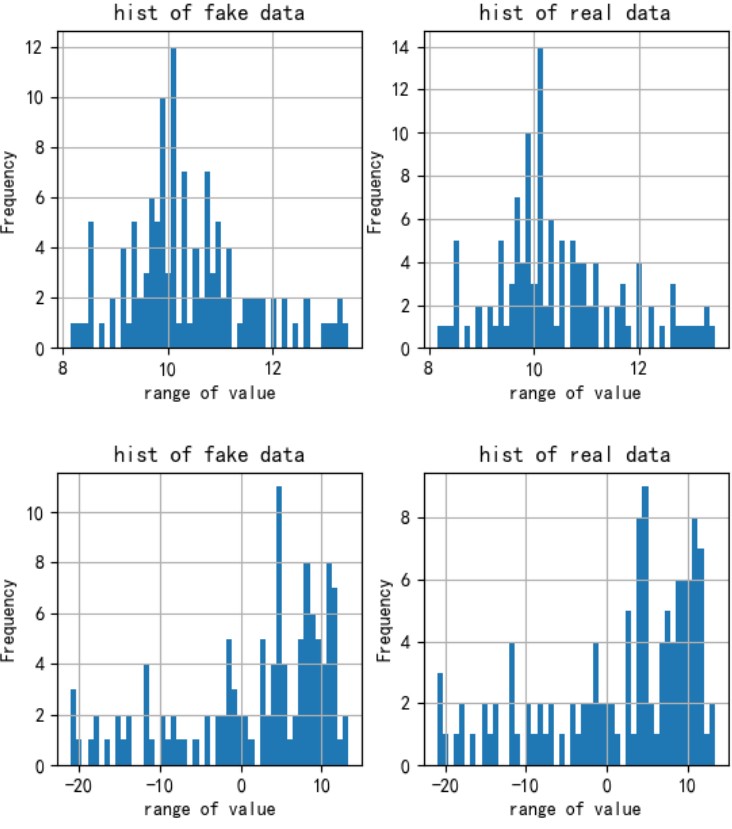

**Figure 7.** Frequency hist of imputation results and the real time series on ETTh1 dataset.

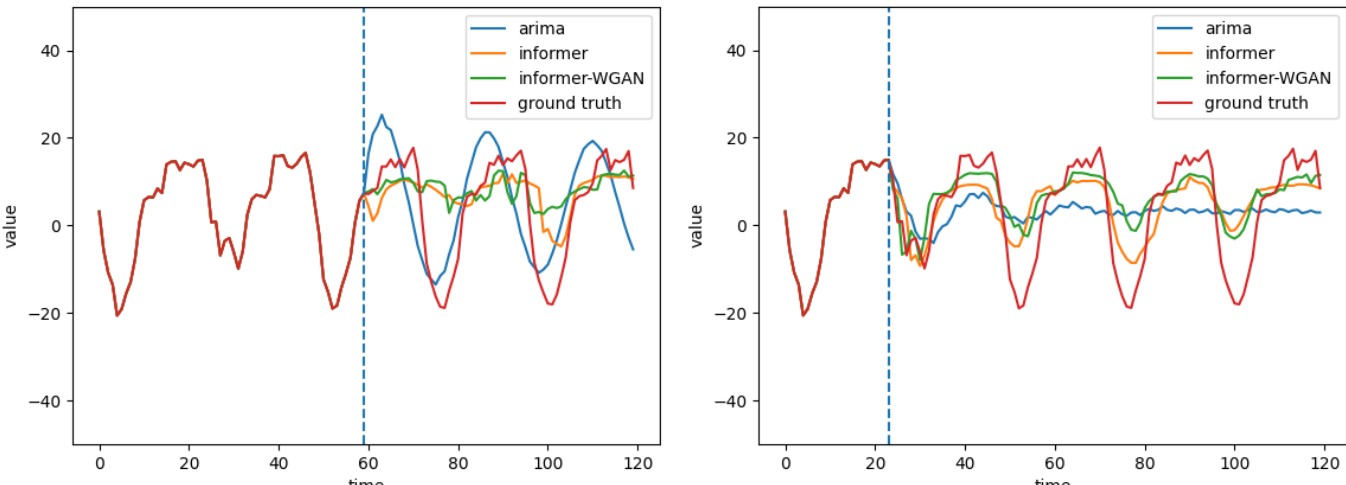

**Figure 8.** Imputation results of different methods on ETTh1 dataset at missing rates of 50% and 80%.

Tables 2 and 3 show the imputation results of our informer-WGAN model on the electricity and ETTh1 datasets. We chose five indicators to evaluate the distance and the Pearson correlation coefficient to evaluate the similarity between the imputaion results and the real data. With the increase in the missing rate, our model demonstrated a slight decrease with respect to all six indicators, which fit our expectation. This means all six indicators can reveal the effect of the imputation task well.

**Table 2.** The results of the informer-WGAN on the electricity dataset.

| Missing Rate | mae | mse | rmse | mape | mspe | p_corr |
|---|---|---|---|---|---|---|
| 20% | 0.82615 | 6.70904 | 2.39374 | 0.19931 | 0.44089 | 0.95462 |
| 40% | 0.86958 | 7.50290 | 2.61123 | 0.21357 | 0.51383 | 0.95277 |
| 60% | 0.89981 | 7.64285 | 2.63168 | 0.23651 | 0.70946 | 0.94944 |
| 80% | 0.97250 | 8.13361 | 2.72510 | 0.27141 | 0.79367 | 0.93473 |

**Table 3.** The results of the informer-WGAN on the ETTh1 dataset.

| Missing Rate | mae | mse | rmse | mape | mspe | p_corr |
|---|---|---|---|---|---|---|
| 20% | 1.90835 | 10.73479 | 3.27281 | 0.53457 | 9.75783 | 0.77216 |
| 40% | 1.97126 | 12.07560 | 3.47173 | 0.54006 | 10.14137 | 0.75321 |
| 60% | 2.01376 | 12.42590 | 3.52390 | 0.58413 | 11.62296 | 0.74823 |
| 80% | 2.05503 | 12.64088 | 3.55464 | 0.57709 | 10.30420 | 0.74734 |

Tables 4 and 5 show the comparison results of our informer-WGAN model and the other four imputation models on two datasets. Our informer-WGAN model showed a better effect with respect to the different indicators.

**Table 4.** The imputation results of different models on electricity dataset at a missing rate of 60%.

| Imputation Model | mae | mse | rmse | mape | mspe | p_corr |
|---|---|---|---|---|---|---|
| ARIMA | 2.47689 | 31.01061 | 4.43682 | 0.20056 | 0.06220 | 0.86310 |
| AST | 2.22961 | 181.73460 | 9.469879 | 1.15476 | 11.27290 | 0.48822 |
| Informer | 1.74750 | 19.67988 | 4.21687 | 0.61075 | 3.34536 | 0.84613 |
| ST-WGAN | 1.35948 | 60.82138 | 5.773326 | 0.86076 | 8.81188 | 0.84930 |
| Informer-WGAN | 0.89969 | 7.64734 | 2.61395 | 0.23624 | 0.70543 | 0.94789 |

**Table 5.** The imputation results of different model on electricity dataset at the missing rate of 60%.

| Imputation Model | mae | mse | rmse | mape | mspe | p_corr |
|---|---|---|---|---|---|---|
| ARIMA | 2.48034 | 19.27467 | 4.35074 | 0.52469 | 4.69485 | 0.65881 |
| AST | 7.87295 | 94.89117 | 9.70722 | 2.44605 | 14.57877 | 0.38196 |
| Informer | 2.11488 | 11.82524 | 3.43685 | 0.59489 | 9.80083 | 0.74467 |
| ST-WGAN | 2.23207 | 21.70739 | 4.60613 | 0.69644 | 24.20782 | 0.65263 |
| Informer-WGAN | 2.01439 | 12.37496 | 3.51492 | 0.56910 | 10.93439 | 0.74725 |

As shown in Tables 6 and 7, the informer-WGAN model we proposed achieved the best results on both the electricity dataset and the ETTh1 dataset. The traditional ARIMA method had poor imputation performance. Compared with the deep learning method, the traditional method had limited utilization of the temporal features, resulting in poor imputation results. However, the accuracy of the AST method decreased significantly when the missing rate increased. Because the sparse transformer model adopted a step-by-step method during decoding (that is, the value of the next time was based on the imputation value of the previous time), it could not be forwarded at one time. The operation was obtained, and the problem of error accumulation occurred. Therefore, the AST model is not suitable for imputation problems with high missing rates. In the informer model, the concept of generative decoder is proposed, and all the imputation was completed through one forward calculation process. The model we proposed also adopts a similar design, which makes the model suitable for imputation under high missing rates. According to the $p\_RMSE$ formula in the previous section, when the quality of the generated sequence is poor, due to the introduction of the Pearson correlation coefficient, the performance gap of the model is more significant.

Tables 8 and 9 show the RMSE and Pearson correlation coefficient of the imputation results and the ground truth on the electricity and ETTh1 datasets. We note that in the table, the ARIMA algorithm obtained the best RMSE value under the 20% missing rate, but its actual imputation result was not close to the true value, so the Pearson correlation coefficient result was poor. The PRMSE value obtained by combining these two statistics was more conducive to obtaining time series imputation results that met the requirements.

**Table 6.** The PRMSE (the smaller, the better) results of the informer-WGAN and other imputation results on electricity dataset.

| Missing Rate | ARIMA | Infomer | Informer-WGAN |
|---|---|---|---|
| 20% | 2.711 | 1.791 | **1.119** |
| 40% | 4.247 | 1.830 | **1.157** |
| 60% | 4.612 | 2.019 | **1.188** |
| 80% | 12.609 | 2.062 | **1.283** |

**Table 7.** The PRMSE (the smaller, the better) results of the informer-WGAN and other imputation results on ETTh1 dataset.

| Missing Rate | ARIMA | AST | Infomer | Informer-WGAN |
|---|---|---|---|---|
| 20% | 2.709 | 2.477 | 2.311 | **2.309** |
| 40% | 3.036 | 7.487 | 2.339 | **2.338** |
| 60% | 3.156 | 9.963 | 2.438 | **2.437** |
| 80% | 3.736 | 19.567 | 2.574 | **2.569** |

**Table 8.** The RMSE/Pearson correlation results of the informer-WGAN and other imputation results on electricity dataset.

| Missing Rate | ARIMA | Infomer | Informer-WGAN |
|:---:|:---:|:---:|:---:|
| 20% | **0.937**/0.345 | 1.337/0.747 | 1.029/**0.920** |
| 40% | 1.081/0.254 | 1.348/0.737 | **1.055**/**0.911** |
| 60% | **0.858**/0.186 | 1.404/0.696 | 1.073/**0.902** |
| 80% | 1.132/0.089 | 1.430/0.694 | **1.130**/**0.880** |

**Table 9.** The RMSE/Pearson correlation results of the informer-WGAN and other imputation results on ETTh1 dataset.

| Missing Rate | ARIMA | Infomer | Informer-WGAN |
|:---:|:---:|:---:|:---:|
| 20% | 2.066/0.762 | 1.795/0.773 | **1.796**/**0.774** |
| 40% | 2.240/0.737 | **1.804**/**0.767** | 1.847/0.756 |
| 60% | 2.311/0.732 | **1.832**/0.750 | 1.867/**0.751** |
| 80% | 2.439/0.652 | 1.903/0.739 | **1.883**/**0.748** |

## 5. Discussion

### 5.1. Performance Comparison

#### 5.1.1. Informer Discriminator vs. Linear Discriminator

In order to verify the influence of our proposed informer decoder-based discriminator on the multi-dimensional time series imputation results, we used the fully connected layer of AST and the informer encoder network we proposed as the discriminator of the model and conducted experiments on the electricity dataset. The experimental results (Table 10) show that when using the informer encoder, the time series imputation at each missing rate was better than using the fully connected layer as the discriminator, which had a significant effect on training a better time series imputation model. The goal of the discriminator is to fit a likelihood function that evaluates the imputation results. When the informer is used as the discriminator, it can make full use of the characteristics of the time series and learn more reasonable evaluation criteria than the linear discriminator. Therefore, the model of the discriminator is the practical choice.

**Table 10.** The ablation experiment results (PRMSE) of different discriminators on electricity dataset.

| Missing Rate | Informer-WGAN (Linear Discriminator) | Informer-WGAN (Informer Discriminator) |
|:---:|:---:|:---:|
| 20% | 1.710 | **1.119** |
| 40% | 1.708 | **1.157** |
| 60% | 1.798 | **1.188** |
| 80% | 1.866 | **1.283** |

#### 5.1.2. Random Training Process

We also conducted studies on a randomized training process to verify the effect of this design on improving model performance. The experimental results are shown in the Table 11. The experiment used for comparison only trained the time series data with a missing rate of 20%. It was foreseeable that the performance of the model under the high missing rate was significantly worse than the random training process. For the informer-WGAN model, as the missing rate increased, the gap was more significant. Not only that, but under the 20% missing rate, the model using the random training process was also significantly better than the model trained with a fixed missing rate. The results show that random training made the model more focused on the time series features, while fixed missing rate training made it easier for the model stuck in a local optimum.

**Table 11.** The ablation experiment results (PRMSE) of different training methods on electricity dataset.

| Missing Rate | Informer-WGAN (20% Train) | Informer-WGAN (Random Train) |
|:---:|:---:|:---:|
| 20% | 1.560 | **1.119** |
| 40% | 2.326 | **1.157** |
| 60% | 2.582 | **1.188** |
| 80% | 2.689 | **1.283** |

## 6. Conclusions

We proposed a new network model for multidimensional time series imputation based on the informer and WGAN methods. We applied the random missing rate training process to train our network in order to obtain a suitable model for the time series imputation task with high missing rates. The results of the imputation experiments on real-world datasets demonstrated the effectiveness of this model in time series imputation, being superior to the ARIMA model, the AST model, and the informer model. At the same time, according to the experimental results, we also verified that the self-attention decoder discriminator was better than the linear discriminator, implying the necessity of the selection of the discriminator model. Our next step is to optimize the informer model used by the generator to improve the running efficiency of the model.

**Author Contributions:** Conceptualization, Y.Q.; methodology, Y.Q.; software, Y.Q.; validation, S.Z. and R.W.; formal analysis, Y.Q.; investigation, Y.Q., S.Z. and R.W.; resources, R.W., L.T. and B.Z.; data curation, Y.Q. and R.W.; writing—original draft preparation, Y.Q.; writing—review and editing, S.Z.; visualization, Y.Q.; supervision, R.W.; project administration, Y.Q.; funding acquisition, L.T. and B.Z. All authors have read and agreed to the published version of the manuscript.

**Funding:** This research was funded by the Civil Aerospace Technology Advance Research Program (A0201) of the State Administration of Science, Technology and Industry for National Defense.

**Institutional Review Board Statement:** Not applicable.

**Informed Consent Statement:** Not applicable.

**Data Availability Statement:** The raw data needed to reproduce these findings cannot be shared at this time, as these data are also part of further research.

**Conflicts of Interest:** The authors declare no conflict of interest.

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
