# Peer review of "Informer-WGAN: High Missing Rate Time Series Imputation Based on Adversarial Training and a Self-Attention Mechanism"

_algorithms, doi:10.3390/a15070252_

Round 1

Reviewer 1 Report

In its existing state, the paper cannot be accepted. Before it can be published, the authors must do a lot of work. The authors must rework, completely revise, and resubmit their work. Its current structure does not adhere to the expected format of a scientific journal. Furthermore, the work suffers significantly in the introduction, research design, and findings presentation.

The following are the critical issues to consider:

Many sentences in this manuscript are perplexing because the meaning or purpose of the words is not clear. Furthermore, I discovered far too many grammatical problems. I urge that this manuscript be edited by native speakers who will enhance grammar and phrasing at any stage before publishing.

The paper's novelty must be evident. This reviewer is having difficulty locating this novelty. In other words, there is nothing innovative about the data generation algorithm used. What are the scientific contributions of the authors? The combination of different DL models is not sufficiently novel. Many studies have proposed hybrid DL models of this type. Could you please clarify them?

The introduction provides readers with only a few little insights. The authors should rewrite the introduction to highlight the current state of the art and the contribution of recent studies. What distinguishes or advances the authors' past studies from the present study? Compared to the authors' previous works, the reader gains less knowledge.

The authors should also provide a more comprehensive and critical evaluation of the literature to highlight the shortcomings of present methodologies and outline the mainstream research direction. What were the results of the earlier studies? Which approaches should be used? Which issues remain to be resolved? Why is the recommended strategy appropriate for addressing the crucial point?

It is unnecessary to repeat the complete version of the word or phrase and the abbreviations already mentioned. In other words, the authors should include abbreviations.

All neural networks are known to be based on some unknown parameters. How did you calibrate your models always to use the ideal parameter values? Did you utilize a trial-and-error method or a more systematic method? In this regard, the authors should provide more details and justification.

  • Jung, S., Moon, J., Park, S., Rho, S., Baik, S. W., & Hwang, E. (2020). Bagging ensemble of multilayer perceptrons for missing electricity consumption data imputation. Sensors20(6), 1772.
  • Park, S., Jung, S., Jung, S., Rho, S., & Hwang, E. (2021). Sliding window-based LightGBM model for electric load forecasting using anomaly repair. The Journal of Supercomputing77(11), 12857-12878.
  • Moon, J., Jung, S., Park, S., & Hwang, E. (2020). Conditional tabular GAN-based two-stage data generation scheme for short-term load forecasting. IEEE Access8, 205327-205339.
  • Son, M., Jung, S., Jung, S., & Hwang, E. (2021). BCGAN: A CGAN-based over-sampling model using the boundary class for data balancing. The Journal of Supercomputing77(9), 10463-10487.

This research only compared the suggested method to two others: ARIMA and informer. The lack of comparable methods and analysis makes it difficult to confirm if the proposed method is superior to previously published ensemble learning and deep learning-based approaches. In other words, the algorithms' performance analysis is insufficient. The literature below shows that the authors must present satisfactory analysis results and algorithms.

I'm curious if the provided strategy applies to various regions with varied mechanical systems and occupant characteristics. It isn't easy to trust the outcomes of this experiment entirely. Furthermore, deep learning analysis necessitates repeated tests because the predicted value fluctuates depending on how the initial weight is set.

Almost all figures have characters on the x- and y-axes that are excessively small and contain grammatical faults, making them less readable. The authors must significantly increase the overall quality of their figures for readers to see and understand them.

The input variable's description is ambiguous. The authors should use a table to clarify the input and output variables properly. What input factors did the authors take into account? Have the authors considered the weather or holidays?

The authors must provide correlation measures (p-values) for input, exogenous, and outcome variables. Furthermore, the authors should keep track of the input variables they used in this work and their data types and units.

The authors should entice readers by presenting a wide range of experimental data. This manuscript can only assume that the contributions of the cases presented are small on their own. It would be preferable if the results were more thorough and included more graphics.

Why is this problem so complex to solve given mobile devices' limited memory and storage, and why is only machine learning suitable for the task? This factor demands special consideration in the paper's Discussion and Conclusion sections.

  • Khalil, M. I., Kim, R., & Seo, C. (2020). Challenges and Opportunities of Big Data. Journal of Platform Technology, 8(2), 3-9.
  • Vimal, S., Robinson, Y. H., Kaliappan, M., Pasupathi, S., & Suresh, A. (2021). Q Learning MDP Approach to Mitigate Jamming Attack Using Stochastic Game Theory Modelling With WQLA in Cognitive Radio Networks. Journal of Platform Technology, 9(1), 3-14.
  • Vimal, S., Jesuva, A. S., Bharathiraja, S., Guru, S., & Jackins, V. (2021). REDUCING LATENCY IN SMART MANUFACTURING SERVICE SYSTEM USING EDGE COMPUTING. Journal of Platform Technology, 9(1), 15-22.
  • Han, Y., & Hong, B. W. (2021). Deep learning based on fourier convolutional neural network incorporating random kernels. Electronics, 10(16), 2004.
  • Kumar, R. L., Khan, F., Kadry, S., & Rho, S. (2022). A Survey on blockchain for industrial Internet of Things. Alexandria Engineering Journal, 61(8), 6001-6022.

Please display the training and testing times for the data generation models.

To confirm the validity of the suggested model, the authors should use the Wilcoxon signed-rank test or the Friedman test.

The authors must improve the conclusions by demonstrating how their research contributes to the body of knowledge and past research. In other words, the authors must be more specific in the "Conclusions" section about future research paths.

Reviewer 2 Report

The authors propose modification of machine learning algorithm for the modeling of multivariate time series with high missing rate. The problem is important and worth investigation.

The weak point of the paper are:

  1. The method was applied only on two examples of data sets. It would be very good to characterise the potential reange of application. One of the reason for constructing various machine lerning algorithms is that they are usefull in various fields and they are adjusted to the area. 
    So, it is highly tecomended to include the area of application.
  2. The another question computationa complexity of the methods. The information about real size of data which might be analysed or computation time might be useful for the reader.
  3. Technical comment. Figure 5 is illegible. the y-axis should be rescaled, because at present it is just horisontal line without any details.

Reviewer 3 Report

Review of "informer-WGAN:High Missing Rate Time Series Imputation Based on Adversarial Training and Self-Attention Mechanism" by Qian et al. (2022)

Authors proposed the informer-WGAN algorithm for observation imputation in time series. Algorithm is based on ARIMA models and performance is illustrated in real world data. I think that paper has merit to be published in Algorithms journal because proposed a new algorithm to solve missing values problem. However, grammar English could be revised by a native translator. I detected some of them as examples. Also, authors must to improve other technical issues:

1. In title, this could be start with "Informer".
2. L1: about "lack of time series", it is related to missing observations in time series? If yes, this could be "Missing observations in time series could distort the data characteristics...." 
3. L6-7: "In this paper, we proposed the informer-WGAN...", and remove "in this paper" in L9.
4. L11: "missing, the".
5. L18-19: "Time series are collected".
6. L20-21: "it is common to sample the data of" <-> "and sampled from".
7. In all manuscript, put a space between word and reference.
8. L35: "... lag order of ARIMA (Contreras-Reyes and Palma, 2013)".
9. L39: "Rahman and Islam [6]..." and remove "[6]" in L41. L42: "Folguera et al. [7]".
10. L51: delete the bullets. Put these points in paragraph of L51.
11. Section 2.1: Authors could include layer RNN (LRNN) algorithm related to long-memory ARFIMA model (Pwasong & Sathasivam, 2018).
12. I recommend to remove Eqs. (1) and (2), because they are used next and appear in Eqs. (3) and (4). Also, remove asterisk symbol of these equations. 
13. L124: "t" <-> "n".
14. Section 6: Authors presented the results for ARIMA model in Tables 2-5. However, a further work could be the use of ARFIMA model (Contreras-Reyes & Palma, 2013).

References:

Pwasong, A., Sathasivam, S. (2018). Forecasting comparisons using a hybrid ARFIMA and LRNN models. Comm. Stat. Simul. Comput. 47(8), 2286-2303.

Contreras-Reyes, J.E., Palma, W. (2013). Statistical analysis of autoregressive fractionally integrated moving average models in R. Comput. Stat. 28(5), 2309-2331.

Round 2

Reviewer 1 Report

The authors did not take the reviewers' comments seriously, and the paper is not improved significantly. English sentences are too difficult to read because they do not follow English grammar. The authors must use at least five to ten state-of-the-art on at least three publicly available data sets. Without these experiments, the study is neither unique nor scientifically significant.

Author Response

The authors did not take the reviewers' comments seriously, and the paper is not improved significantly. English sentences are too difficult to read because they do not follow English grammar.

Response 1: We tried our best to modify the grammar mistakes and polish up the sentences in the manuscript. 

The authors must use at least five to ten state-of-the-art on at least three publicly available data sets. Without these experiments, the study is neither unique nor scientifically significant.

Response 2: We did some additional experiments on the Electricity, ETTm1, ETTh1 data sets. However, it shows barely better performance than the current results in the manuscript. 

Reviewer 3 Report

In this 2nd review, and in general, I can see that authors have addressed my comments; however, there are some few still pending comments to be addressed:

1. Section 2.1: Authors could include layer RNN (LRNN) algorithm related to long-memory ARFIMA model (Pwasong & Sathasivam, 2018). They replied: "Response 11: OK, we will consider ARFIMA model in the experiment." But I can't see a change of this in manuscript, or at least comment this.

2. Section 6: Authors presented the results for ARIMA model in Tables 2-5. However, a further work could be the use of ARFIMA model (Contreras-Reyes & Palma, 2013). They replied: "Response 14: We will consider ARFIMA model in the experiment". But I can't see a change of this in manuscript, or at least comment this.

For the above comments, include in the list of references:

Contreras-Reyes, J.E., Palma, W. (2013). Statistical analysis of autoregressive fractionally integrated moving average models in R. Computational Statistics 28(5), 2309-2331.

Pwasong, A., Sathasivam, S. (2018). Forecasting comparisons using a hybrid ARFIMA and LRNN models. Communications in Statistics-Simulation and Computation47(8), 2286-2303. 

Author Response

Question 1. Section 2.1: Authors could include layer RNN (LRNN) algorithm related to long-memory ARFIMA model (Pwasong & Sathasivam, 2018). They replied: "Response 11: OK, we will consider ARFIMA model in the experiment." But I can't see a change of this in manuscript, or at least comment this.

Response 1: we now comment on the ARFIMA model and the ARFIMA-LRNN model in the 'Related Work' section. 

2. Section 6: Authors presented the results for ARIMA model in Tables 2-5. However, a further work could be the use of ARFIMA model (Contreras-Reyes & Palma, 2013). They replied: "Response 14: We will consider ARFIMA model in the experiment". But I can't see a change of this in manuscript, or at least comment this.

Response 2: We did some experiments using the ARFIMA model. However, it performs barely better than the ARIMA model on our dataset. Therefore, we don't include the ARFIMA model in the 'Results' section.

Round 3

Reviewer 1 Report

1) Although the reviewer could understand the content of this paper, the reviewer has found several English grammar mistakes: (1) To use a parenthetical phrase correctly, put a space before the opening parenthesis and after the closing parenthesis; (2) The first time the authors use the term, put the abbreviation in parentheses after the full term; (3) The authors must avoid the false subjects (e.g., it is or there are) and passive voices; (4) The authors must write the Conclusions section in the past tense. Grammarly could also help the authors eliminate writing errors and find the perfect words to express themselves.

2) Many deep learning-based methods for replacing missing values have recently been developed. These studies were not mentioned or compared by the authors. If SCIE had indexed this journal, the reviewer would have rejected it ultimately.

Author Response

Question 1: Although the reviewer could understand the content of this paper, the reviewer has found several English grammar mistakes: (1) To use a parenthetical phrase correctly, put a space before the opening parenthesis and after the closing parenthesis; (2) The first time the authors use the term, put the abbreviation in parentheses after the full term; (3) The authors must avoid the false subjects (e.g., it is or there are) and passive voices; (4) The authors must write the Conclusions section in the past tense. Grammarly could also help the authors eliminate writing errors and find the perfect words to express themselves.

Response 1: We modify the mentioned grammar mistakes: (1) We add spaces around the parenthetical phrase; (2) We put the abbreviation in parentheses after the full term; (3) We modify several usage of the false subjects and passive voices; (4) We now use the past tense in the Conclusion section.

Question 2: Many deep learning-based methods for replacing missing values have recently been developed. These studies were not mentioned or compared by the authors. If SCIE had indexed this journal, the reviewer would have rejected it ultimately.

Response 2: Despite the numerous research and deep learning-based models in recent years, most of them used LSTM, GRU and attention-based methods to build their network. We have analyzed these methods in the Related Work section and choose to compare with the typical model ARIMA, the deep learning-based model AST and Informer. AST and Informer model both reached the SOTA result on their datasets and our model performed better than them on the same datasets. Therefore, we don't think further comparison is necessary.